# RadiOnCOVID: Multicentric Analysis of the Impact of COVID-19 on Patients Undergoing Radiation Therapy in Italy

**DOI:** 10.3390/cancers17010086

**Published:** 2024-12-30

**Authors:** Andrea Emanuele Guerini, Giulia Marvaso, Sandro Tonoli, Giulia Corrao, Maria Ausilia Teriaca, Matteo Sepulcri, Melissa Scricciolo, Alessandro Gava, Sabrina Montrone, Niccolò Giaj-Levra, Barbara Noris Chiorda, Giovanna Mantello, Francesco Fiorica, Simona Borghesi, Liliana Belgioia, Angela Caroli, Alba Fiorentino, Barbara Alicja Jereczek-Fossa, Stefano Maria Magrini, Michela Buglione

**Affiliations:** 1Department of Radiation Oncology, Istituto del Radio O. Alberti, Spedali Civili Hospital, Piazzale Spedali Civili 1, 25121 Brescia, Italy; stefano.magrini@unibs.it (S.M.M.); michela.buglione@unibs.it (M.B.); 2Department of Radiation Oncology, Università degli Studi di Brescia, 25123 Brescia, Italy; 3Division of Radiation Oncology, European Institute of Oncology IRCCS, 20141 Milan, Italy; giulia.marvaso@ieo.it (G.M.); giulia.corrao@ieo.it (G.C.); barbara.jereczek@ieo.it (B.A.J.-F.); 4Department of Oncology and Hemato-Oncology, University of Milan, 20122 Milan, Italy; 5UOC Radioterapia e Medicina Nucleare, ASST di Cremona, 26100 Cremona, Italy; sandro.tonoli@asst-cremona.it; 6Radiotherapy and Radiosurgery Department, Humanitas Research Hospital, 20089 Rozzano, Italy; maria.ausilia.teriaca@cancercenter.humanitas.it; 7Radiotherapy, Veneto Institute of Oncology, IOV-IRCCS Padua, 35128 Padova, Italy; matteo.sepulcri@iov.veneto.it; 8Radiation Oncology Division, Clinical Oncology Department, Ospedale dell’Angelo, 30174 Venice, Italy; melissa.scricciolo@aulss3.veneto.it; 9Department of Radiation Oncology, Azienda Ospedaliera ULSS 9, 31100 Treviso, Italy; alessandro.gava@aulss2.veneto.it; 10U.O. Radioterapia, Azienda Ospedaliero-Universitaria Pisana, Via Roma 69, 56120 Pisa, Italy; sabri84na@gmail.com; 11Dipartimento di Radioterapia Oncologica Avanzata, IRCCS Sacro Cuore Don Calabria Hospital, Via Don Angelo Sempreboni, 37124 Negrar di Valpolicella, Italy; niccolo.giajlevra@sacrocuore.it; 12Department of Radiation Oncology, Fondazione IRCCS Istituto Nazionale dei Tumori, 20133 Milan, Italy; barbara.noris@istitutotumori.mi.it; 13SOD Radioterapia (Dipartimento Medicina Interna), Azienda Ospedaliero Universitaria delle Marche, 60002 Ancona, Italy; giovanna.mantello@ospedaliriuniti.marche.it; 14Dipartimento di Oncologia Clinica, AULSS 9 Scaligera, 37122 Verona, Italy; francesco.fiorica@aulss9.veneto.it; 15Radiation Oncology Unit of Arezzo-Valdarno, Azienda USL Toscana Sud Est, 52100 Arezzo, Italy; simona.borghesi@uslsudest.toscana.it; 16U.O. Radioterapia Oncologica, IRCCS Ospedale Policlinico San Martino, 16132 Genova, Italy; liliana.belgioia@unige.it; 17Dipartimento di Scienze della Salute (DISSAL), Università Degli Studi di Genova, 16126 Genova, Italy; 18SOC Oncologia Radioterapica, Centro di Riferimento Oncologico, Istituto di Ricovero e Cura a Carattere Scientifico, 33081 Aviano, Italy; angela.caroli@cro.it; 19Department of Medicine and Surgery, LUM University, 70010 Casamassima, Italy; fiorentino@lum.it; 20Department of Radiation Oncology, Generale Regional Hospital F Miulli, 70021 Acquaviva delle Fonti, Italy

**Keywords:** COVID-19, coronavirus, radiotherapy, radiation therapy

## Abstract

A study conducted across 19 Italian Radiation Oncology centers examined 41,039 cancer patients undergoing or scheduled for anticancer treatment between February 2020 and May 2021. The study aimed to assess the impact of COVID-19 on treatment schedules and patient outcomes. Among the cohort, 123 patients were diagnosed with COVID-19 during treatment (group A), and 99 were diagnosed before treatment initiation (group B). The overall COVID-19 incidence was 0.54%, including 0.30% in group A. Severe COVID-19 developed in 60 patients, and 45 died due to the infection (incidence of 0.15% and 0.11%, respectively). Treatment delays or suspensions were common, with 37.4% of group A requiring temporary suspension, 32.5% definitive suspension and 37 patients continuing treatment while positive. In group B, 53.5% faced delays, and 20.2% had definitive suspensions. Most of the patients with a COVID-19 diagnosis in our cohort recovered and completed their treatment; nonetheless, the attributable death rate after confirmed infection was 20.27%. With adequate preventive measures, radiation oncology could continue safely during the pandemic, resulting in a low incidence of severe outcomes and mortality.

## 1. Introduction

Coronavirus disease 2019 (COVID-19) had an overwhelming effect on healthcare worldwide, requiring an unprecedented effort of the medical community to face the pandemic and its consequences.

The impact has been even more evident for patients requiring anticancer therapy, as treatment delay might have a major impact on clinical outcomes. Moreover, some previous experiences suggest that oncologic patients might be at higher risk of developing severe forms of COVID-19 [1,2,3,4].

Presentation of COVID-19 is frequently mild or asymptomatic and may overlap with symptoms associated with cancer (e.g., lung cancer) or antineoplastic treatment (e.g., immunotherapy), thus hindering a prompt diagnosis [5].

Several radiation oncology guidelines provided suggestions with the aim of offering lifesaving and urgent treatment while limiting the chance of acquiring SARS-CoV-2 infection [6,7,8].

As Italy was the first Western Nation involved in the pandemic, during its ‘first wave’, few, if any, clinical data were available, and thus, the management of patients requiring radiotherapy (RT) was initially heterogeneous and led, in some instances, to a significant reduction in the activity of several Centers [9].

Since the beginning of the pandemic, the Italian Association of Radiation Oncology (Associazione Italiana di Radioterapia ed Oncologia Clinica—AIRO) advised against treatment omission or suspension, at least for the subset of patients for which postponement or discontinuation could be detrimental in terms of disease control [10].

A thorough re-organization of the activity and the adoption of dedicated preventive measures allowed the continuation of treatments and follow-up visits [11,12,13].

Despite the global spread of COVID-19 and the increasing information on its outcomes in neoplastic patients, literature regarding the effects of the pandemic on subjects undergoing RT is still limited and mostly represented by single-center experiences [14].

We thus implemented a cooperative multi-center investigation, including a retrospective analysis followed by a prospective phase, to evaluate the impact of COVID-19 on Italian radiation oncology institutions and their patients undergoing active anticancer treatment.

## 2. Methods and Materials

Data of all the patients older than 18 years who received active antineoplastic treatments (including radiotherapy and systemic therapy) at the participating institutions between 3 February and 31 December 2020 (retrospective phase) and between 1 January and 31 May 2021 (observational prospective phase) were analyzed. COVID-19 diagnosis was defined according to World Health Organization criteria as the positivity of a Nucleic Acid Amplification Test (NAAT) test on a nasopharyngeal swab performed by trained personnel. Primary outcomes included incidence of SARS-CoV-2 infection (defined as the number of confirmed cases divided by the number of patients on active treatment in the index period), COVID-19-related mortality and number of severe presentations, definition of risk factors associated with death and severe COVID-19 and impact of COVID-19 on antineoplastic treatment (e.g., delay or suspension). Pseudo-anonymized data were collected using a specifically developed website, including an online database, assessing demographic and clinical features including sex, age, cancer type and stage, Eastern Cooperative Oncology Group Performance Status (ECOG-PS), comorbidities, Body Mass Index (BMI) and smoking habit. Severity of COVID-19 was classified as follows: (a) asymptomatic in case of positive NAAT without any sign or symptom suggestive of COVID-19; (b) mild in presence of sign or symptoms consistent with COVID-19 (e.g., fever, fatigue, alteration of smell and taste, headache, intestinal symptoms) without dyspnea and/or other criteria defining severe disease; (c) severe in case of dyspnea, reduction of oxygen saturation measured by pulse oximetry (SpO2) below 94% (or reduction of more than 5% compared with baseline), respiratory rate over 30 breaths/min, septic conditions, shock, organ failure, or any condition secondary to COVID-19 requiring hospital admission. Antineoplastic treatment characteristics and timing and outcomes of COVID-19 and its impact on radiotherapy or systemic therapy were described. Treatment suspension was defined as temporary if treatment was continued and completed after discontinuation and as definitive if treatment was permanently discontinued. Follow up was performed according to the usual clinical practice of each Institution. The database was formatted and processed using IBM-SPSS^®^ software (version 25.1). Categorical variables were presented as frequencies or percentages, and continuous variables presented as means or medians and compared with the use of Student’s *t*-test, Anova or the Mann–Whitney and Kruskal–Wallis test; statistical significance was set at a *p*-value α ≤ 0.05. The protocol was conducted according to Good Clinical Practice (GCP) and the Declaration of Helsinki. The study was approved by the Ethics Committee of ASST Spedali Civili di Brescia (number of approval NP 4316).

## 3. Results

### 3.1. COVID-19 Incidence Among Patients with Active or Planned Anticancer Treatment

A total of 41,039 patients were treated or had a planned treatment at one of the 19 Italian Radiation Oncology departments included in the study between 1 February 2020 and 31 May 2020.

Overall, 343 patients were affected by the COVID-19 pandemic, including: 123 patients with NAAT-confirmed diagnosis during active treatment (group A), 99 patients with confirmed diagnosis before antineoplastic treatment start (group B), 50 subjects with a presentation suggestive of COVID-19 during treatment not confirmed by NAAT, 35 cases with a presentation suggestive of COVID-19 before treatment with negative NAAT, 23 patients who experienced treatment delay due to quarantine after contact with a COVID-19 case, 5 patients that refused hospital access in fear of contagion and 8 patients that had logistic problems due to the pandemic.

The characteristics of patients from groups A + B are summarized in Table 1.

The total incidence of COVID-19 across patients with active or planned treatment in this period was 0.54% (groups A + B) and 0.30%, considering only patients with a positive NAAT performed while already on active treatment (group A). Considering the sum of group A and group B, a total of 60 patients developed severe COVID-19 and a total of 45 patients died as a consequence of the infection, leading to an incidence of 0.15% and a mortality of 0.11%, respectively. In group A, 34 severe cases and 29 deaths were reported (incidence of 0.08% and 0.07%).

The number and incidence of total and severe COVID-19 cases, as well as deaths likely caused by COVID-19 divided by month, are summarized in Table 2. While the peak of cases substantially overlapped in March 2020 and November 2020 (1.42% vs. 1.43%), the incidence of severe cases considerably reduced over time (0.65% in March 2020 vs. 0.27% in November 2020 and 0.16% in January 2021) and similarly, mortality declined (0.49% in March 2020 vs. 0.15% in November 2020 and 0.16% in January 2021).

This trend is even more evident comparing the three COVID-19 waves in Italy during the study period (Table 2). The incidence of total and severe COVID-19 cases, as well as deaths likely due to COVID-19, in patients with active or planned treatment, were 0.45%, 0.21% and 0.16%, respectively, during the first wave (February–May 2020), 0.81%, 0.15% and 0.09% in the second wave (October 2020—January 2021) and 0.28%, 0.06% and 0.04% amidst the third wave (February–May 2021).

### 3.2. COVID-19 Outcomes

Considering the entire group of 222 patients with a confirmed COVID-19 diagnosis, severity was reported for 215 subjects: 69 cases (32.1%) were asymptomatic, 86 were mild (40%) and 60 cases (27.9%) were severe. Hospitalization was required in 39 cases (median duration 22 days), and death attributable to COVID-19 was reported for 45 cases (20.27%).

Taking into account the 123 subjects diagnosed with COVID-19 during active treatment (data regarding severity available for 121 patients, median dose at diagnosis 24 Gy), 37 cases were asymptomatic, 50 mild and 34 severe (respectively 30.6%, 41.3% and 28.1%). In 28 cases, patients were hospitalized, and 29 subjects (23.58%) died as a consequence of COVID-19. Chest X-ray was performed in 77 cases (of which 50.6% had suggestive findings), and CT scan in 46 subjects (positive in 60.9% of the cases). At the last assessment, COVID-19 negative testing was reported for 75 cases (60.98%), while 38 cases (30.89%) were still positive, and 10 cases (8.13%) were not assessable.

Of the 99 patients with a COVID-19-positive test before treatment start, severity was reported for 94: 32 (34%) were asymptomatic, 36 (38.3%) were mild and 26 (27.7%) were severe. Negative testing was reported for 73 cases (73.7%) at last assessment, while 16 (16.2%) were still positive and 10 not assessable. Death likely due to COVID-19 was reported in 16 subjects (16.16%).

### 3.3. Impact of COVID-19 on Anticancer Treatment

Among the 123 patients with COVID-19 diagnosis while already on active therapy, 46 cases (37.40%) required temporary treatment suspension (median and mean duration 20 days), 40 cases (32.52%) definitive suspension and 37 patients did not necessitate suspension and continued treatment while positive. Median RT-prescribed dose was 45 Gy, and median dose per fraction was 2.5 Gy; in 25 patients (20.33%), the dose prescription was modified in order to complete the treatment. No toxicity increase likely associated with COVID-19 was described.

As for the 99 patients with a COVID-19 diagnosis before treatment start, 53 (53.53%) experienced temporary delay (median 18.5 days, mean 25.6 days), 20 (20.20%) definitive treatment suspension and 26 (26.26%) no delay; in 9 subjects (9.09%), RT was performed while positive.

Patients who underwent RT while COVID-19 positive did not develop unexpected or increased toxicities.

### 3.4. Risk Factors for Treatment Suspension, Severe Disease and Death

Older age was significantly associated with definitive treatment suspension (*p* = 0.02, median age 73.8 years in case of definitive suspension versus 67.3 years in case of temporary suspension and 64 years in case of no suspension), with severe COVID-19 (*p* = 0.000, median age 62.2 years for asymptomatic infection versus 64.8 years for subjects with mild symptoms and 73.1 years for patients with severe COVID-19) and with death likely due to COVID-19 (median age 75.2 years for deceased versus 64.2 for non-deceased patients).

An ECOG PS > 2 was significantly associated with definitive treatment suspension (*p* 0.005), severe COVID-19 (*p* 0.001) and death likely due to COVID-19 (*p* 0.000).

Male sex was associated with definitive treatment suspension (*p* 0.005), severe COVID-19 (*p* 0.006) and death likely due to COVID-19 (*p* 0.021).

The site of the tumor was not associated with treatment suspension (*p* 0.076), with the exception of lung cancer, which was well associated with severe COVID-19 (*p* 0.001) and with a non-significant trend toward death likely due to COVID-19 (*p* 0.056).

Administration of systemic therapy was not associated with treatment suspension (*p* 0.258), COVID-19 severity (*p* 0.788) and death likely due to COVID-19 (*p* 0.955).

## 4. Discussion

The coronavirus disease pandemic had a devastating effect on cancer screening, diagnosis and treatment.

Despite the growing number of publications and guidelines in the oncology setting, only a limited number of clinical experiences have been published about patients with COVID-19 undergoing RT, mostly represented by retrospective single-center series [15,16,17]. Updated data from two previously published retrospective studies [11,14] were included in this analysis. To the best of our knowledge, this is the largest cohort of adequately described confirmed COVID-19 patients undergoing radiation therapy.

A similar experience was published by Ots et al. [17]: among the 39,848 patients registered in 66 Spanish radiation oncology centers (February–May 2020), 329 COVID-19 cases were identified (incidence 0.8%); nonetheless, the proportion of patients undergoing active treatment was not specified, complete data were available for 235 subjects and only 214 patients performed NAAT (positive in 146 cases).

Data regarding mortality (22.1%) and treatment modifications (70.6%) were consistent with our findings, although a lower proportion of patients completed the planned treatment (50.2%).

Previous series regarding COVID-19 in cancer patients [18,19,20,21] reported heterogeneous incidences (0.38–7.8%) that were in line with our results (total incidence 0.54%), although generally higher. This could be due to the definition of COVID-19 case, which, in our experience, required a positive NAAT, while in previous studies, it could also include radiological or clinical diagnosis.

A trend could be observed over the three COVID-19 “waves” covered by our observation period: despite the higher incidence reported in the second wave compared with the first wave (0.81% vs. 0.45%), the incidence of severe cases (0.15% vs. 0.21%) and mortality (0.09% vs. 0.16%) were lower.

This might be partly explained by the increasing knowledge available to clinicians and the fact that, while initially there was a lack of NAAT availability and only patients with suggestive symptoms or at risk of exposure were tested, in following months, a larger proportion of patients was tested, and surveillance performing nasopharyngeal tests to a larger proportion of the patients resulted in relatively high incidence of COVID-19 infection, but with reduced rates of severe presentation [22].

The third wave presented lower incidence and further reduction of serious cases and deaths. These findings could be connected with the start of the vaccine campaign against COVID-19 in December 2020 [23] that, as described in the literature, limited COVID-19 incidence and consequent hospitalization and death [24,25,26] and proved to be safe in cancer patients [27], although antibody and neutralization titers could be lower in subjects receiving radiotherapy [28].

This finding is in line with previous reports that demonstrated reduced COVID-19 severity and mortality between the different waves in cancer patients [29].

Prognosis after infection was dismal in our series, with (considering group A + B) 27.9% of patients presenting severe disease and a mortality of 20.27% (that reached 23.6% in patients diagnosed during active treatment). Mortality rates aligned with those reported in previous series evaluating cancer patients (10.9–38%) [21,30,31,32,33]. This highlights the importance of adequate COVID-19 prevention, screening and diagnosis in patients during antineoplastic treatment.

We identified older age (*p* 0.000), ECOG PS > 2 (*p* 0.001), male sex (*p* 0.006) and lung cancer (*p* 0.001) as risk factors for severe COVID-19, and older age (*p* 0.000), ECOG PS > 2 (*p* 0.000) and male sex (*p* 0.021) as risk factors for death due to COVID-19. In previous cancer patient series, older age [4,21,31,32,34], male sex [4,31,32] and lung cancer were similarly identified as risk factors for worse prognosis after COVID-19 infection.

Therefore, a group of patients at higher risk could be identified and should be considered for more intensive screening and prophylactic programs.

It should be noted that the total incidence of severe COVID-19 (0.15%) and death likely due to COVID-19 (0.11%) over the whole observation period in our cohort were remarkably low, and mortality was even lower than the one observed across the Italian population during the index period (0.22%) [35]. Moreover, as highlighted above, both severe disease and death likely due to COVID-19 markedly decreased over time.

It should also be considered that the majority of COVID-19-positive patients completed planned treatment: 67.5% taking into account patients infected during treatment (37.4% with temporary suspension) and 79.8% considering subjects infected awaiting treatment start (53.5% with delay). Therefore, the rate of definitive treatment suspension due to COVID-19 among the whole cohort was limited (60 cases, 0.15%).

Risk factors for definitive treatment suspension in our cohort were older age (*p* 0.020), an ECOG PS > 2 (*p* 0.005) and male sex (*p* 0.005).

Taking into account the 46 patients who continued radiotherapy while COVID-19 positive, no events of increased toxicity were reported.

Treatment interruption allows neoplastic cell repopulation [36] with a detrimental effect on disease control [37], and previous experiences demonstrated the deleterious effects of radiotherapy delay and suspension due to COVID-19 [38].

Considering the above-mentioned data, treatment continuation or start in patients diagnosed with COVID-19 should be considered on the basis of clinical presentation and risk factors following a personalized framework in order to minimize the hazard of compromising cancer control. Although heterogeneous policies have been adopted across the institutions participating in this study, most of the centers endorsed AIRO recommendations against treatment postponement or discontinuation through the implementation of dedicated preventive measures.

The limits of this study should also be acknowledged, including the retrospective collection of the data for the first months (leading to the loss of some relevant information) and the frequent difficulty in collecting complete data during the first pandemic outbreak. The incidence of COVID-19 in our population, although coherent with previous reports, was relatively low. This could be due to the stringent criteria adopted for COVID-19 diagnosis. Moreover, this study did not specifically analyze the impact of cancer-related outcomes in the few patients who experienced treatment delays due to COVID-19. Finally, the unceasingly changing landscape of the COVID-19 impact on healthcare makes our results only partially applicable to future cohorts of patients.

## 5. Conclusions

As in the previous series, mortality was elevated for cancer patients diagnosed with COVID-19 during active treatment. Nonetheless, the overall incidence of COVID-19 was low, with severe presentations and mortality rates being even lower and remarkably reduced over time. The majority of patients that acquired infection completed planned treatment, and across the whole cohort, only 0.15% of patients required definitive treatment suspension due to COVID-19. Therefore, with the implementation of adequate preventive measures, radiation oncology activities could safely continue during the COVID-19 pandemic.

## Figures and Tables

**Table 1 cancers-17-00086-t001:** Characteristics of the 222 patients with active or planned treatment and confirmed COVID-19 diagnosis.

		All	
		Median	Mean
Age (years)		72.17	-
Number of comorbidities		1	1.53
BMI		25	25.37


		N	%
Sex	Male	128	57.66
	Female	92	41.44
	Not specified	2	1
Type of cancer	Breast cancer	42	18.91
	Prostate cancer	32	14.41
	Lung cancer	26	11.71
	Gastro-intestinal cancer	17	7.66
	Brain cancer	16	7.21
	Gynaecologic cancer	16	7.21
	Not specified	73	32.88
Stage	Stage I	35	15.77
	Stage II	26	11.71
	Stage III	47	21.17
	Stage IV	83	37.39
	Not specified	31	13.96
ECOG performance status	PS 0	91	41.00
	PS 1	74	33.33
	PS 2	26	11.71
	PS 4	10	4.5
	Not specified	21	9.46
Smoking status	Non smoker	103	46.4
	Former smoker (cessation more than two years before radiotherapy)	40	18.02
	Current smoker	68	30.63
	Unknown status	11	4.95
Systemic treatment	No	118	53.15
	Concomitant	82	36.94
	Exclusive	17	7.66
	Not reported	5	2.25

	Chemotherapy	54	24.32
	Hormone treatment	33	14.86
	Immunotherapy	3	1.35
Radiotherapy indication	Radical	58	26.12
	Adjuvant	69	31.08
	Ablative	18	8.11
	Palliative	55	24.77
	Neoadjuvant	9	4.05
	Other	7	3.15
Radiotherapy site	Pelvic	45	20.27
	Breast	36	16.21
	Thorax	34	15.32
	Head and neck	28	12.61
	Bone	26	11.71
	Brain	24	10.81
	Upper-gastrointestinal	5	2.25
	Lower-gastrointestinal	3	2.25

ECOG = Eastern Cooperative Oncology Group; BMI = Body Mass Index.

**Table 2 cancers-17-00086-t002:** Number and incidence of total and severe COVID-19 cases and death likely due to COVID-19.

	Pts Treated	Group A Positive	Group A Severe	Group A Deaths	Group A + B Positive	Group A + B Severe	Group A + B Deaths
Feb 2020	2605	0	0	0	0	0	0
Mar 2020	2459	22 (0.89%)	15 (0.61%)	11 (0.45%)	35 (1.42%)	16 (0.65%)	12 (0.49%)
Apr 2020	2274	4 (0.18%)	1 (0.04%)	0	6 (0.26%)	3 (0.13%)	4 (0.18%)
May 2020	2378	0	0	0	2 (0.08%)	1 (0.04%)	0
Jun 2020	2632	0	0	2 (0.08%)	1 (0.04%)	1 (0.04%)	2 (0.08%)
Jul 2020	2775	0	0	0	0	0	0
Aug 2020	2373	0	0	0	1 (0.04%)	1 (0.04%)	0
Set 2020	2670	0	0	0	1 (0.04%)	0	0
Oct 2020	2656	21 (0.79%)	3 (0.11%)	0	28 (1.05%)	4 (0.15%)	0
Nov 2020	2590	19 (0.73%)	6 (0.23%)	3 (0.12%)	37 (1.43%)	7 (0.27%)	4 (0.15%)
Dec 2020	2420	12 (0.50%)	3 (0.12%)	2 (0.08%)	24 (0.99%)	3 (0.12%)	3 (0.12%)

Jan 2021	2501	4 (0.16%)	0	3 (0.12%)	7 (0.28%)	4 (0.16%)	4 (0.16%)
Feb 2021	2481	1 (0.04%)	0	0	6 (0.24%)	1 (0.04%)	0
Mar 2021	2847	13 (0.46%)	2 (0.07%)	0	15 (0.53%)	2 (0.07%)	0
Apr 2021	2658	3 (0.11%)	2 (0.07%)	1 (0.04%)	5 (0.19%)	2 (0.08%)	2 (0.08%)
May 2021	2720	1 (0.04%)	1 (0.04%)	1 (0.04%)	3 (0.11%)	1 (0.04%)	1 (0.04%)

First wave	9716	27 (0.28%)	16 (0.16%)	11(0.11%)	44 (0.45%)	20 (0.21%)	16 (0.16%)
Second wave	12,648	57 (0.45%)	12 (0.09%)	8 (0.06%)	102 (0.81%)	19 (0.15%)	11 (0.09%)
Third wave	8225	17 (0.21%)	5 (0.06%)	2 (0.02%)	23 (0.28%)	5 (0.06%)	3 (0.04%)

Total	41,039	123 (0.30%)	34 (0.08%)	29 (0.07%)	222 (0.54%)	60 (0.15%)	45 (0.11%)

First wave: February–May 2020; second wave: October 2020–January 2021; third wave: February–May 2021. Classification of the severity was presented in ‘Methods’ section. Pts = patients.

## Data Availability

The data presented in this study are available on request from the corresponding author.

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
