# Peer review of "RadiOnCOVID: Multicentric Analysis of the Impact of COVID-19 on Patients Undergoing Radiation Therapy in Italy"

_cancers, 2024, doi:10.3390/cancers17010086_

Round 1

Reviewer 1 Report

Comments and Suggestions for Authors

Proposed paper is interesting and well written. However, some revisions are needed before it can be accepted for publication:

- Are there any differences between COVID waves? Numerical data have been reported without p-values. In fact, it was already demonstrated (as an example: 10.1007/s10389-021-01675-y) that mortality and complications were very different between the different waves. 

- Could authors give also information on how Italian RT outpatients services overcome the impossibility of patients visit in the first wave? there were an increase in cancer related outcomes due to logistical delay? 

Author Response

Proposed paper is interesting and well written. However, some revisions are needed before it can be accepted for publication:

We would like to thank the Reviewer for the insightful suggestions

- Are there any differences between COVID waves? Numerical data have been reported without p-values. In fact, it was already demonstrated (as an example:10.1007/s10389-021-01675-y) that mortality and complications were very different between the different waves. 

Considered the differet situation among the three COVID-19 waves and the evaluation of ther incidence in the three timepoints, only absolute spread among the incidence during the three waves could be calculated, as chi-square test could not be performed and therefore p-value could not be identified. Nonetheless, we added a paragraph and included the valuable reference provided.

- Could authors give also information on how Italian RT outpatients services overcome the impossibility of patients visit in the first wave? there were an increase in cancer related outcomes due to logistical delay? 

Most of the Institution endorsed AIRO reccommendations to avoid treatment delay and postponement due to COVID-19. Nonetheless, a few patients still experienced treatment delay. Although this could be of a lesser magnitude considered the entire cohort, the impact on this patients was not analyzed specifically. We added this limit of the study in the Discussion section.

Reviewer 2 Report

Comments and Suggestions for Authors

The present multicenter cohort study presents interesting retrospectively collected data regarding the impact of COVID pandemic on oncological patients treated with Radiotherapy.

Major concerns that have to be addressed include:

1. English language editing for overall quality improvement

2. Please use the STROBE checklist for cohort studies throughout the manuscript

3. The use of a flowchart will augment the scientific soundness of the manuscript

4. Tables are too extensive please make them more short and clear.

Comments on the Quality of English Language

English language editing is needed throughout the text

Author Response

REVIEWER 2.

The present multicenter cohort study presents interesting retrospectively collected data regarding the impact of COVID pandemic on oncological patients treated with Radiotherapy.

Major concerns that have to be addressed include:

We would like to thank the Reviewer for the detailed analysis of the manuscript and insightful suggestions.

1. English language editing for overall quality improvement

A thorough editing of the text has been performed, with the aid of a native English speaker.

2 and 3. Please use the STROBE checklist for cohort studies throughout the manuscript. The use of a flowchart will augment the scientific soundness of the manuscript

We checked the manuscript according to STROBE checklist and added missing items, such as follow up method and definition of treatment suspension. Moreover we included additional limitations of this study.

4. Tables are too extensive please make them more short and clear.

Most of the data included in the Tables are necessary to support our results and complete the manuscript.

Reviewer 3 Report

Comments and Suggestions for Authors

The study presented in the manuscript is compelling. The objectives are clearly defined, and the methodologies employed are suitable. All pertinent information is thoroughly detailed.

The study's findings hold significance from two perspectives: (1) societal, regarding the delivery of radiotherapy services during the pandemic, and (2) individual, regarding the outcomes for patients undergoing radiotherapy who have developed COVID-19. Both topics are effectively explored in the manuscript.

However, the second aspect, although critical (with a COVID-19 attributable death rate of 20.27%), is somewhat downplayed, as it is not mentioned in the Simple Summary and Abstract. I would suggest that this oversight be corrected.

Author Response

The study presented in the manuscript is compelling. The objectives are clearly defined, and the methodologies employed are suitable. All pertinent information is thoroughly detailed.

The study's findings hold significance from two perspectives: (1) societal, regarding the delivery of radiotherapy services during the pandemic, and (2) individual, regarding the outcomes for patients undergoing radiotherapy who have developed COVID-19. Both topics are effectively explored in the manuscript.

However, the second aspect, although critical (with a COVID-19 attributable death rate of 20.27%), is somewhat downplayed, as it is not mentioned in the Simple Summary and Abstract. I would suggest that this oversight be corrected.

We would like to thank the Reviewer for the appreciation of our work.

The insightful suggestions provided have been implemented.

Reviewer 4 Report

Comments and Suggestions for Authors

Dear authors, 

Please find my comments below.

This paper statistically examines the impact of the COVID-19 pandemic on radiotherapy in Italy using a national oncology registration database.

The impact of COVID-19 on cancer treatment is significant and therefore the current study should be of interest to readers of Cancers.

However, it needs to be revised in several respects before it can be considered for publication in the journal.

1. There is a lack of proper punctuation throughout the text. There are many words and phrases without commas, making the whole sentence difficult to understand. For example, 'few if any' in l57. A grammar check by a native English speaker is highly recommended to improve the readability of the paper.

2. The term "definitive suspension" in l171 should be clearly defined in the methods section. Also, "exclusive" under the systemic therapy row in Table 1 and "increased toxicity" in l180 should be explained (e.g.; a criterion for the latter should be given).

3. Table 1; there is a significant decrease in the number of positive cases from May 2020 to September 2020. Is this due to the social closure during this period?

4. Abstract; in the conclusion, the expression "the majority recovered" should be corrected, because a significant proportion of patients (45 out of 60 cases) with severe COVID-19 infection died of the infection.

Comments on the Quality of English Language

A grammar check by a native English speaker is highly recommended to improve the readability of the paper.

Author Response

This paper statistically examines the impact of the COVID-19 pandemic on radiotherapy in Italy using a national oncology registration database.

The impact of COVID-19 on cancer treatment is significant and therefore the current study should be of interest to readers of Cancers.

However, it needs to be revised in several respects before it can be considered for publication in the journal.

1. There is a lack of proper punctuation throughout the text. There are many words and phrases without commas, making the whole sentence difficult to understand. For example, 'few if any' in l57. A grammar check by a native English speaker is highly recommended to improve the readability of the paper.

We would like to thank the reviewer for this valuable suggestion. A thorough editing of the text has been performed, with the aid of a native English speaker.

2. The term "definitive suspension" in l171 should be clearly defined in the methods section. Also, "exclusive" under the systemic therapy row in Table 1 and "increased toxicity" in l180 should be explained (e.g.; a criterion for the latter should be given).

The required definitions have been implemented in the text for clarity.

Treatment suspension was defined as temporary if treatment was continued and completed after discontinuation, and as definitive if treatment was permanently discontinued.

3. Table 1; there is a significant decrease in the number of positive cases from May 2020 to September 2020. Is this due to the social closure during this period?

As correctly pointed out, the decrease of positive cases between May and September 2020 is due to the end of the first COVID-19 wave, likely facilitated by social closure.

4. Abstract; in the conclusion, the expression "the majority recovered" should be corrected, because a significant proportion of patients (45 out of 60 cases) with severe COVID-19 infection died of the infection.

We thank the reviewer for this insightful correction, that has been implemented in the text.

Round 2

Reviewer 2 Report

Comments and Suggestions for Authors

Authors made an effort to adress most of the comments. However, I still suggest the revision of the tables as they are very extensive and not so necessary.

Comments on the Quality of English Language

Minor improvements can be done.

Author Response

We would like to thank the Reviewer for the comments, that allowed us to improve the quality of the manuscript. The few redundant elements in the Tables were removed, while we believe that other information included are relevant for the manuscript.

Reviewer 4 Report

Comments and Suggestions for Authors

The authors have corrected the manuscript properly according to the comments.

Author Response

We would like to thank the Reviewer for the valuable insights, that allowed us to improve the quality of the manuscript.